# Comparing Residents' Fear of Crime with Recorded Crime Data—Case Study of Ostrava, Czech Republic

**Jiří Pánek** [1,*] 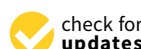**, Igor Ivan** [2] **and Lucie Macková** [1]

1   Department of Development and Environmental Studies, Palacky University Olomouc,
    771 46 Olomouc, Czech Republic
2   Department of Geoinformatics, VSB—Technical University of Ostrava, 708 00 Ostrava, Czech Republic
*   Correspondence: jiri.panek@upol.cz

**Abstract:** The fear of crime is an established research topic, not only in sociology, environmental psychology and criminology, but also in GIScience. Using spatial analysis to analyse patterns, explore hotspots and determine the significance of respective surveys is one reason for the increase in popularity of such research topics for geographers, cartographers and spatial data scientists. This paper presents the results of an intensive online map-based questionnaire with 1551 respondents from the city of Ostrava, Czech Republic. The respondents marked 3792 points associated with the fear of crime over a ten week period. The perception data were compared with recorded crime data acquired from police department records for the years 2015–2018. This paper explores the spatial autocorrelation from perceived hotspots and from recorded crime hotspots. Our findings fit into the literature confirming results about the locations that most frequently attract fear, but there is still room for more investigations regarding the links between recorded crime and the fear of crime.

**Keywords:** VGI; participatory mapping; crime mapping; PPGIS

## 1. Introduction

Understanding people's responses to the fear of crime [1] is important for understanding behaviours which reflect this concern. The fear of crime is one of the essential concepts of Criminology, and is often described as an emotional response to potential victimisation [2]. Historically, the fear of crime was defined as the likelihood of the risk [3,4]. In other words, how likely it is to become a victim of a crime. However, recent studies [5,6] link the fear of crime to a person's emotional response. Moreover, it is important for policymakers and municipalities, who are able to act upon these concerns. Importantly, the fear of crime affects individuals and their well-being, and it may even influence their behaviour as regards to frequenting the respective sites.

On the other hand, there has been a vast amount of research focusing purely on analysing and exploring patterns in recorded crime data [7–19] relating to the time of day and land use. This research combines the perceived crime data with recorded crime data in one analysis in order to explore the attributes of various parts of the selected city—Ostrava, Czech Republic.

Geographic information systems (GIS) are well-positioned to integrate the data on people's perceptions of the urban environment with police crime reports. Using GIS can enable the police and local communities to address the fear of crime and recorded crime in a targeted manner. The residents' perceptions of crime might not be reflected in reported crimes and so the police could have a new tool with which to pinpoint underreported criminal activity [20] and to improve prevention communication in areas where crime is underestimated.

This research focuses on the Czech city of Ostrava. The city was selected for this case study because it was involved in the project titled "Effective Methods of Identification, Assessment and

Monitoring of Safety Risk Areas Using Spatial Micro-data" run by the VSB-Technical University of Ostrava. Involvement in the project is linked with prior collaboration with this participating university and their willingness to share crime occurrence data and other indicators that would help to identify safety risk areas in general. The city of Ostrava was selected for this case study following almost one-year's preparation and cooperation in the area of perception of fear mapping with the local administration. The city was selected based on the authors previous experience with participatory mapping, the knowledge of local realities and the second worst crime intensity in the country (after Prague). Another reason for selecting the city for this case study was the availability of recorded crime data for Ostrava, as this was the first time such data were released for any large city in the Czech Republic. Ostrava is the third largest city in the Czech Republic and it is located close to the north east border with Poland. It was founded in 1267 and it is a coal-mining city. The wider conurbation of the region has a population of approximately 500,000. Since the 1990s, after the rapid decline of the city's industrial sectors such as iron, steel and coal-mining, Ostrava has been transformed into a modern cultural city with numerous theatres, galleries and other cultural facilities.

The long-term aim of the project is to compare the subjective/emotional fear of crime perception with reported crime incidents. A comparison of such combined data has never been done in the Czech Republic prior to this research. Hence, this gives a unique insight into the Czech fear of crime/recorded crime landscape. The aim of the paper is to investigate whether the fear of crime occurs in the areas where the recorded crime is and vice versa. The data are not compared with the population density, as such, the data do not exist for the Czech Republic. There are data of where people live, but this is a density of sleeping people, and it is quite different from real-time population density in the city.

From September to November 2018, an online participatory mapping survey similar to [21] was launched focusing on the perceptions of the residents of selected cities as regards to safety and crime. The survey design was created by the authors and the municipal representatives, and included mapping the sites where people felt unsafe, the reasons why and the possible actions taken by the residents. Drawing from the tradition of environmental psychology and building on the experience of participatory mapping, the authors analyse the results with the aim of: (1) Identifying statistically significant hotspots from the responses in each city; (2) analysing whether there are any similarities in these hotspots within the cities; (3) comparing perceived crime hotspots with recorded crime hotspots. The paper is structured as follows: (1) A literature review on fear of crime introduces the research topic; (2) the methods used for data collection are presented; (3) the results of the case study of Ostrava are outlined and followed by (4) a discussion concerning the comparison of our results with other research. The paper ends with a short conclusion that may be read as an invitation for future research collaboration in the area of the fear of crime analysis. The raw data collected are available for browsing at the link provided towards the end of the paper. The data from the police are not public and therefore, cannot be browsed or shared.

## 2. Background

Research has identified crime and anti-social behaviour as important influences on the residents' well-being [22,23]. The fear of crime is a major dimension in the formation of the quality of life, yet it is often disregarded from a public policy standpoint [24]. Many studies have suggested a link between the fear of crime, social disorder and serious crime (e.g., [25]), although "broken windows" theory was criticised to focus only on the physical and forgetting about social factors influencing the fear of crime [26]. The examples of this critique can be found in [27]. The fear of crime emerged as a stand-alone concept in the 1960s, but it has been defined and measured in different ways [28–30]. The fear of crime is often understood as a negative emotional reaction demonstrated as the fear of criminality [3] or victimization [31]. In this paper, the authors understand the fear of crime as an emotional response towards places in a city where respondents do not feel safe, in accordance with the main question in the questionnaire (see more in Methods and Results): "Mark the places where you feel unsafe". The fear of crime can be perceived differently by different social groups (including

age, gender, ethnicity, class and disability), and this has been discussed in various studies [32,33]. Curtis [34] asserted that representing feelings or beliefs about the urban environment can enable the understanding of their impact on behaviour. The studies by critical and feminist scholars have contributed new insights to the understanding of the emotions related to the perception of fear and crime in the urban environment [35–37]. It has been found that victimisation (being previously a victim of crime) and physical vulnerability, such as age or gender, greatly influence the fear of crime [38].

Research also shows that specific types of crime correlate with particular characteristics of urban spaces [39]. Therefore, it is important to consider these aspects when planning urban design. For example, Stankevice et al. [40] demonstrated that specialised areas and greenery in dense residential areas contribute to crime prevention. However, if these areas are combined with local centres, commercial or industrial areas, they become even more attractive to criminals. Similarly, Hillier and Sahbaz [41] argued that mixed-use street segments are relatively safe in the majority of cases and that an increased residential population neutralises the risks associated with sparse residence in mixed-use areas. Moreover, the lack of residents in public spaces and the discontinuity of public spaces increase crime because of the lack of oversight by other residents. This is in line with the pioneering findings of Jacobs [42], who claimed that urban spaces with mixed use lead to less crime because they provide more natural surveillance. The studies also show [7] that crime hotspots can change over time, but the papers also discuss the spatial periodicity of the crime itself [43].

Crime and the fear of crime can be concentrated in hotspots, which can have direct effects on the experiences of pedestrians [44]. The different types of land use might attract different types of crime [7]. Monteiro [45] found that areas with a concentration of trade tend to attract robbery and burglary. Sypion-Dutkowska and Leitner [10] found that certain areas, such as alcohol outlets, clubs and discos, cultural facilities and municipal housing, attracted crime. Moreover, commercial buildings also indirectly attracted crime. The strong influence of land use was limited to within 50 m of the surrounding area. Railways, bus stations and other public transportation nodes may also be perceived as less safe because they can encourage crimes where target density is crucial, such as pickpocketing or mugging [46]. As has been argued by Clarke [47], the fear of crime that prevents many from using public transport has a serious impact on revenues. Research from Mexico City found that around one-third of public transportation users felt unsafe or not very safe, and that public transportation is an important, albeit neglected, dimension of policies targeting the quality of life [24]. Matijosaitiene et al. stated that in Manhattan, New York, the open spaces and outdoor recreational areas generate larceny. This usually happens in periods of darkness due to the lower numbers of visitors, and during the day because of the larger number of strangers. As with the transportation hubs, the presence of strangers in urban spaces makes them less safe [7]. Thus far, most of the literature compares the fear of crime locations within one city with their respective land use, or different times of the day, or proximity to various points of interest (transportation hubs, parks, pubs, open spaces, etc.). Some research compares the fear of crime in different neighbourhoods of one city [48], but the comparison of the fear of crime with recorded crime data has been quite limited [49].

## 3. Methods and Results

The data collection was carried out by the researchers from September to November 2018 using social networks as well as official channels of the city administration. A web-based digital map (see Figure 1) designed by the authors as a simple crowdsourcing webpage was used to collect the data. The language of the survey was Czech (the webpage was translated into English only for the needs of this paper). For the data collection, the points were identified as the optimal feature class, and beside sketching polygons [50], the use of points was the predominant method for spatially-explicit preference mapping. The authors preferred the points over polygons as they tend to be easier for respondents to grasp and usually have a higher completion rate in the mapping activity [51], although it is necessary to have a higher amount of answers. On the other hand, polygon areas outside hotspots may represent "potential spatial errors from using polygons" ([51] p. 239).

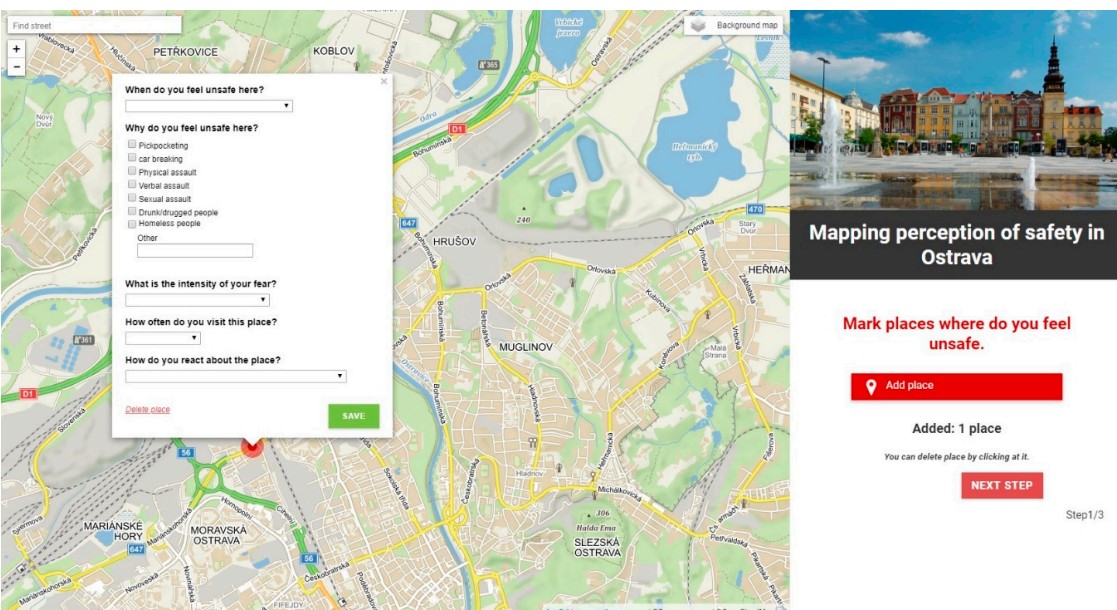

**Figure 1.** Print screen of the mapping application interface used for data collection.

In this research, there was only one spatial question—*Where do you feel unsafe?* This was followed by a fairly intensive questionnaire with non-spatial questions regarding the times of day respondents do not feel safe, the reasons for their feelings, how often they visit places where they feel unsafe and how they react to their unpleasant feelings. Furthermore, the demographics were collected concerning age, gender, the neighbourhood where they live and how safe they rate their neighbourhood compared to other areas of the city. Nevertheless, this paper analyses only the spatial part of the questionnaire.

In total, the data set has 1551 respondents, who marked 3792 points during the ten week data collection period. Gender distribution is more or less balanced with a slight over-representation of women (51.6%). On the other hand, the age distribution does not fully reflect the normal distribution as Ostrava has several universities with a large number of students. Nevertheless, the previous research about emotional mapping [52] proved that there are no significant differences between the answers of university students and the general population. Whether this is the case in Ostrava has not yet been proved.

Most respondents felt unsafe (Figure 2) at the main train station, which is near the city centre and also surrounded by low income housing and derelict buildings. Similarly, people felt unsafe at another train station (Ostrava-Svinov), which is in the north-west of the city. The main street in Ostrava´s neighbourhood, Poruba was also perceived as unsafe. In the city centre, people felt unsafe in the areas leading from the train station to the Forum Nová Karolina shopping centre and along the famous Stodolní street, which has many bars and pubs. The southern part of the city had several key hotspots where people felt unsafe—among them, the Ostrava-Vítkovice and Ostrava-Hrabůvka areas. This may have mainly been caused by cheap local hostels and localities with a higher occurrence of drunks and drug addicted people.

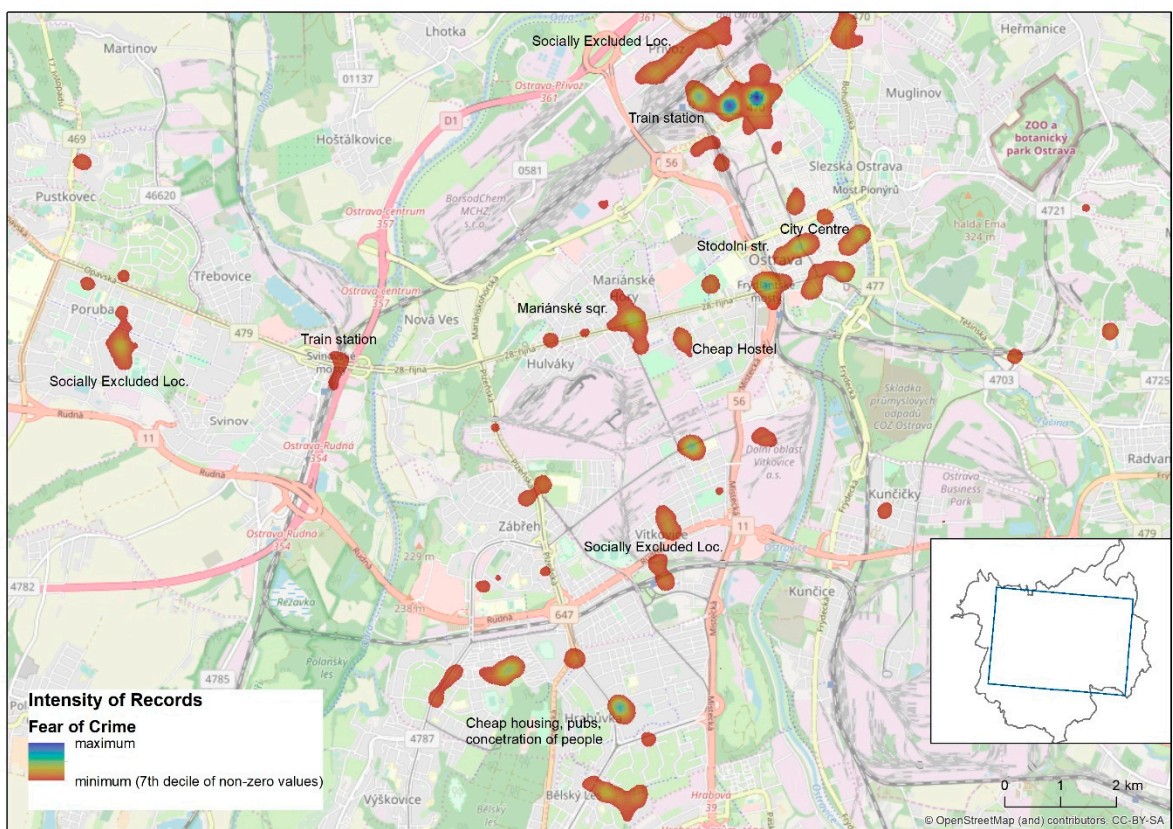

**Figure 2.** Perception of unsafe places in Ostrava.

### 3.1. Recorded Crime Data

The map based crime statistics are in Europe and often are available only on a country or regional level [53]. In the Czech Republic, the most detailed resolution of recorded crime data available so far were police districts. In Ostrava, it would be nine districts. For this research, the recorded crime data was provided by the Police Presidium of the Czech Republic. The data covered the period from January 2015 to July 2018 and contained all recorded crime offences which occurred in Ostrava and within a 5 km buffer area beyond its borders. There were 257,381 incidents in total. Each incident was classified according to a crime offence category, class and subclass, and localised in space using geographical coordinates, and in time. The pre-processing of data consisted of filtering out crime offences that were localised at police stations because of unknown actual localisation (e.g., during travel on a tram), and this applied to 4% of all the data.

There were several reasons for the selection of the five categories/classes of criminal offences (violent crime, burglaries, extremism, personal theft and car robbery). Firstly, these crime offences were recommended for use in this study by the police and the Ostrava authorities. Secondly, the total number of incidents in each of the selected categories/classes were high enough for further spatial analysis, compared to many other categories/classes which had a limited number of incidents (e.g., sexual offences, arson, fraud). Thirdly, the spatial distribution of events is not influenced to the same extent by external factors as some other categories/classes (e.g., traffic accidents, driving offences). Fourthly, the selected categories/classes can be compared with the categories defined in the fear of crime data.

### 3.2. Spatial Analytics

To analyse the spatial distribution of crime and the fear of crime data, the method of spatial autocorrelation was used to detect how the incidents tend to be clustered in space. All events were aggregated to a hexagonal grid with the hexagon sides 40 m long. This side was selected after a

sensitivity analysis comparing Moran's I for hexagon sides from 20 to 120 m (with 20 m steps). Table 1 is comparing Moran's I, z-value (using 999 random permutations) and the number of hexagons. It is evident that the Moran's I value remains very similar (from 0.28 to 0.35) with the increasing hexagon side. The highest Moran' s I is for 20 and 40 m long hexagon side, while the total number of hexagons for 40 m side is four times smaller than 20 m side, which is what supports the selection of 40 m long hexagon sides.

**Table 1.** Change of Moran's *I* of the fear of crime data with increasing length of the hexagon side.

| Hexagon Side | Moran's *I* | z-Value | N of Hexagons |
|---|---|---|---|
| 20 | 0.35 | 737.20 | 1,217,705 |
| 40 | 0.35 | 301.04 | 314,944 |
| 60 | 0.33 | 177.86 | 135,660 |
| 80 | 0.28 | 146.95 | 76,500 |
| 100 | 0.31 | 112.78 | 49,200 |
| 120 | 0.33 | 114.23 | 34,200 |

Table 2 summarises the global spatial autocorrelation using Moran's *I* [54] and using queen adjacency that is defined as:

$$I = \left[ \frac{n}{\sum_i^n (y_i - \overline{y})^2} \right] \times \left[ \frac{\sum_i^n \sum_j^n w_{ij}(y_i - \overline{y})(y_j - \overline{y})}{\left( \sum_i^n \sum_j^n w_{ij} \right)} \right] \tag{1}$$

where *i* and *j* refer to different hexagons, *y* is the data value in each, $\overline{y}$ represents the overall mean, $w_{ij}$ represents an element from the spatial weights matrix (equal to 1 if hexagons share the border or 0 if hexagons do not share the border). For more details, see, e.g., [55].

**Table 2.** Moran's *I* of recorded crime data and the fear of crime data.

| Type | Category | Moran's *I* | z-Value |
|---|---|---|---|
| Registered Crime | Violent Crime | 0.11 | 105.86 |
| | Burglary | 0.11 | 101.19 |
| | Car Robbery | 0.25 | 245.34 |
| | Personal Theft | 0.14 | 224.80 |
| | Extremism | 0.06 | 55.31 |
| Fear of Crime | Personal Theft | 0.30 | 284.82 |
| | Car Robbery | 0.20 | 206.29 |
| | Assault | 0.26 | 241.79 |
| | Verbal Attack | 0.30 | 269.07 |
| | Sexual Attack | 0.18 | 187.14 |
| | Offense by drunk/drug person | 0.32 | 257.58 |
| | Offense by homeless person | 0.33 | 260.32 |
| | All events | 0.35 | 301.04 |

All Moran's *I* values are statistically significant (see z-values in Table 2 using 999 random permutations), although the values are rather small because many hexagons are without any event. The strongest autocorrelation from the recorded crime data is in the case of car robbery, while the other four categories have a very similar level of spatial autocorrelation. The fear of crime data show typically higher spatial autocorrelation with Moran's *I* of all events equal to 0.35. The category with

the strongest autocorrelation is the fear of offences committed by drunk or homeless people, with 0.32 and 0.33 respectively.

To find areas in the city with a higher concentration of recorded crime events or the fear of crime events, a local spatial autocorrelation was used. A cluster map (sample in Figure 3) was created for all the above-selected categories of the crime data and the fear of crime data, in order to augment the significant locations with an indication of the type of spatial association [56]. Each hexagon in these cluster maps was classified into one of five categories:

- High-high (dark red)—high value (number of events) in the hexagon is surrounded by high values (number of events) in surrounding hexagons relative to the overall mean;
- Low-low (dark blue)—low value in the hexagon is surrounded by low values in surrounding hexagons relative to the overall mean;
- High-low (light red)—high value in the hexagon is surrounded by low values in surrounding hexagons relative to the overall mean;
- Low-high (light blue)—low value in the hexagon is surrounded by high values in surrounding hexagons relative to the overall mean;
- Not-significant 0.75 the result is not statistically significant.

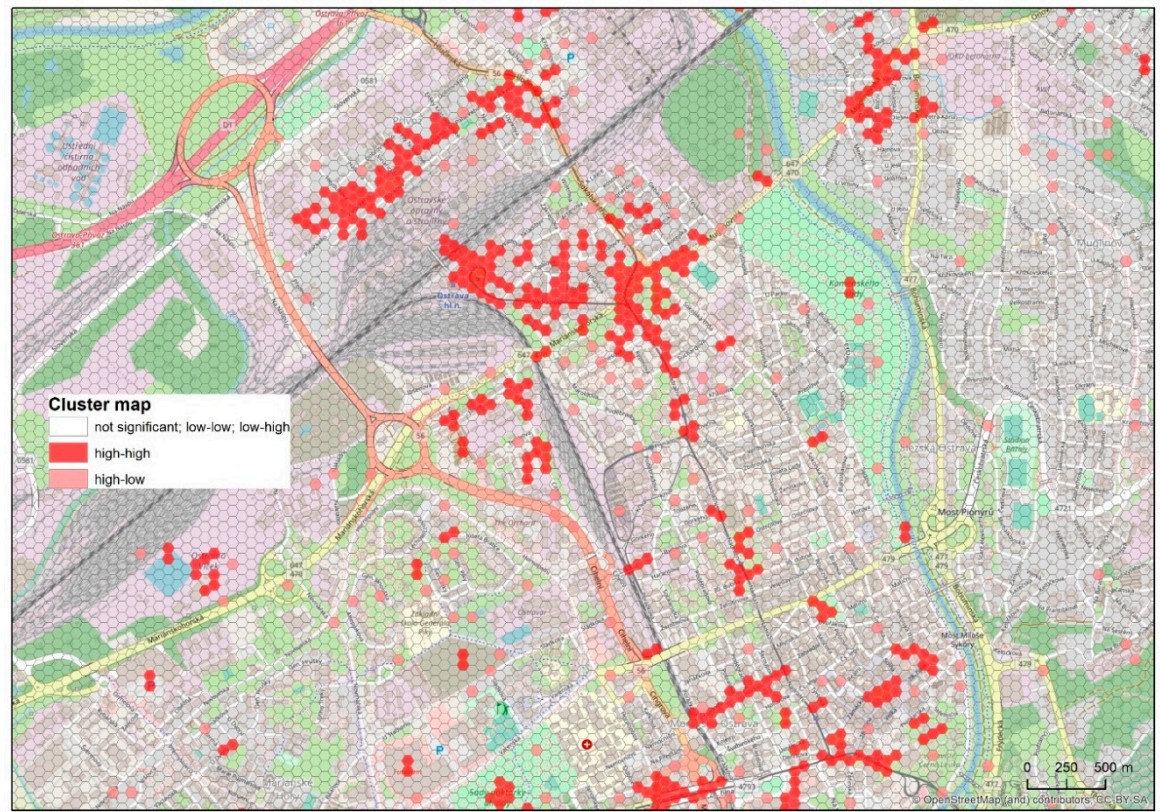

**Figure 3.** Cluster map of fear of crime events (all events).

Hexagons classified as members of high-high clusters represent the core areas with a significantly higher number of events and high-low cluster highlights the locations with spatial outliers (higher number of events) compared to surrounding hexagons (see Figure 3).

A combination of two different cluster maps was used to compare the locations with higher intensities of crime and the fear of crime and to find areas where the fear of crime is justified by a higher intensity of recorded crime (Figure 4). Each cluster map classified all hexagons into five different categories, so the total number of combinations for the two cluster maps was 25 categories. In order to

make the final visualisation comprehensive, categories were organised into nine resulting groups. The most important category is the combination of high-high × high-high clusters because these represent the only hexagons with a significantly high level of recorded crime and the fear of crime in the hexagon itself and in the adjacent hexagons (description in Table 3). All hexagons that were not classified into a high-high cluster in any of the two combined cluster maps were grouped into one class that was not displayed on the final maps. These hexagons represent the locations with a low or no significant intensity of events (recorded crime and fear of crime). The only exception is the combination of high-low × high-low classes where a hexagon contains a higher number of events (recorded crime and fear of crime) compared to the overall mean value. While there were only a few members of this class, they were grouped with the high-high × high-high class. The remaining combinations of a high-high cluster in one cluster map and an insignificant, low-low, low-high or high-low in the second cluster map make up the other eight categories. These eight categories can be divided into two groups, depending on whether the recorded crime or the fear of crime (hues of green were used in the maps below) belongs to the high-high cluster. In cases where the fear of crime belonged to the high-high cluster, the hues of blue were used in the maps below. These hexagons represent the locations where the fear of crime is not justified by the recorded crime data. In contrast, the green hues highlight hexagons with a high-high cluster membership in recorded crime data only. At these locations, people did not declare a higher level of the fear of crime despite the significantly high level of criminal activity. The hue of blue or green corresponds to the intensity of the non-high-high variable (fear of crime or recorded crime).

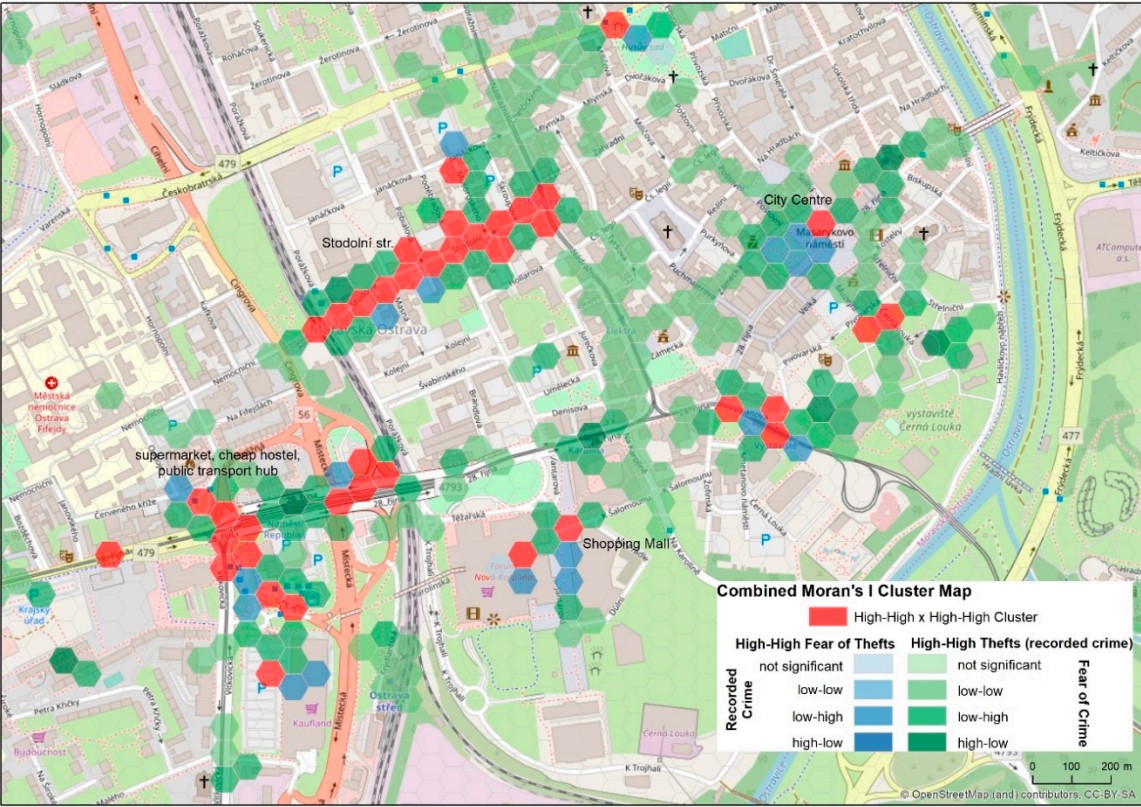

**Figure 4.** Cluster map comparing the fear of personal thefts and thefts as recorded crime incidents.

**Table 3.** The description of classes used in a combined cluster map.

| Combinations (Fear of Crime × Recorded Crime) | Description |
| --- | --- |
| high-high × high-high | High level of fear in the hexagon and adjacent hexagons (compared to the overall mean value value) and high level of crime intensity in the hexagon and in adjacent hexagons (compared to the overall mean value) |
| high-high × not significant | High level of fear in the hexagon and adjacent hexagons (compared to the overall mean value value) and not a significant level of crime intensity in the hexagon and in adjacent hexagons (compared to the overall mean value) |
| high-high × low-low | High level of fear in the hexagon and adjacent hexagons (compared to the overall mean value) and low level of crime intensity in the hexagon and in adjacent hexagons (compared to the overall mean value) |
| high-high × low-high | High level of fear in the hexagon and adjacent hexagons (compared to the overall mean value) and low crime intensity in the hexagon together with high intensity of crime in adjacent hexagons (compared to the overall mean value) |
| high-high × high-low | High level of fear in the hexagon and adjacent hexagons (compared to the overall mean value) and high crime intensity in the hexagon together with low intensity of crime in adjacent hexagons (compared to the overall mean value) |

The combined cluster map comparing the fear of personal thefts, with thefts as recorded crime incidents shows that the city centre of Ostrava has two high crime density areas. The first is along Stodolní street, famous for its large number of pubs and concentration of young (and often drunk) people during Friday and Saturday nights. The second place highlighted in red, is the area of the public transport hub (combination of tram, trolleybus and train) with a supermarket, a cheap hostel, and a hazardous area below the bridge with a high occurrence of aggressive individuals or groups. In general, the green hues predominate on the map, and this is caused by a significantly higher number of recorded crime events than the selected locations based on the questionnaires. Hexagons with a darker green or blue colour are evidently closer to red hexagons, and the light green colour represents hexagons with a very small number of events.

The combinations of the different types of crime and the fear of crime were also tested, and their correlations were proved. The highest correlation from all combinations ($r = 0.22$) is for the combination of fear of being attacked by a drunk or drug addicted person(s) and personal theft (recorded crime). In the eastern part of the map (Figure 5), two locations are visible, that were previously described, but have a higher number of red hexagons. Another hot spot area is in the western part of the map (Figure 5), around Mariánské square, an area with the presence of socially excluded persons. A larger area with an unjustified fear of crime can be found in the central part of the map. The main reason for the increased level of fear is a cheap local hostel.

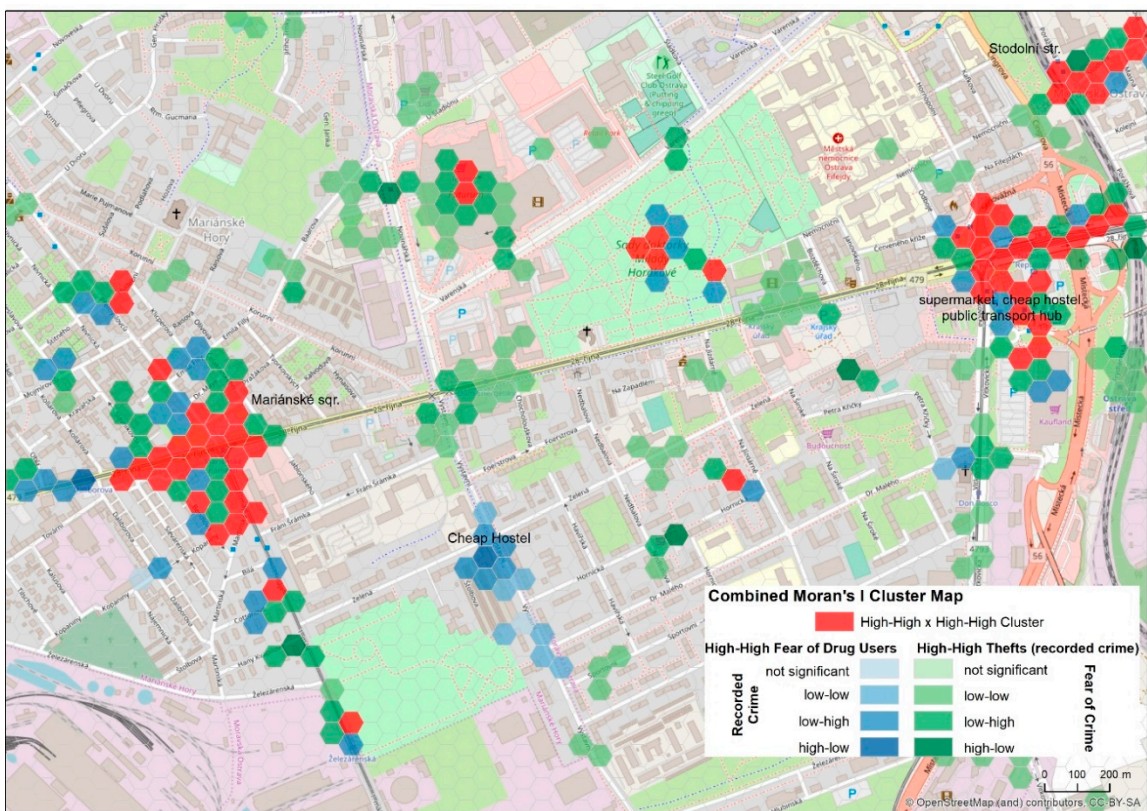

**Figure 5.** Cluster map comparing fear of being attacked by a drunk or drug addicted person(s) and personal thefts (recorded crime).

## 4. Discussion

The allocation of hotspots in relation to the perception of safety (Figure 3) did not provide many surprises, and the results are in accordance with the results from other fear of crime research, not only in the Czech Republic [52,57,58], but also in other countries [59–61]. The usual suspects are transportation hubs, in this case they are train/bus stations, urban parks (mainly during the night), shadowy narrow streets with limited public lighting and/or public spaces in close proximity to nightclubs/bars/gambling houses.

Gender is considered one of the most important predictors of the fear of crime, as women fear crime at higher levels than men [1,5]. This is usually explained by their higher sensitivity to risk and also that women perceive the consequences of risk more seriously than men [59]. This is particularly interesting when taking into consideration that women are less likely to become victims of crime and yet they fear crime more [1,62]. When asked if they had been victims of a criminal act in the last 12 months, 22% of the respondents replied that they had, but almost half of those did not notify the police—which means that half the criminal acts in our data sample were unreported! In this case study, the results confirm the research stated above by (a) women marking more (by 21%) places as unsafe and (b) less women being victims of crime than men in the past 12 months.

Similar to the retrospective question about being a victim in the past 12 months, the respondents were asked about their perception of changes in the safety in their neighbourhood over the previous 24 months. While 49% of respondents saw no change, 40% of respondents believed that the situation had worsened or significantly worsened. Only 11% saw improvement or significant improvement in their perception of safety. This is also reflected in their response to the question of whether they were aware of any activities that the city was organising in the area of crime prevention where 67% of respondents answered negatively.

Another aspect of this research was the question of how people react to the fear of crime, whether they proactively protect themselves or change their behaviour somehow. Regarding the constraints behaviour, the literature divides the constraints into (a) protective constraints (owning a weapon, having a guard dog, taking up a self-defence classes, etc.) and (b) avoidance constraints (avoiding certain places, only going out in groups, etc.) [1,63]. Our results show that 35% of the respondents do not take any protective action, while 33% are on the side of avoidance constraints (21% avoid places and 12% always walk in groups). On the other hand, 20% of the respondents declared that they take the protective path, and 12% answered other, which may be a combination of both approaches. The respondents' protective behaviour also depends on the intensity of the fear of crime assigned to each place. While in the locations marked with a small intensity of the fear of crime, the respondents mostly (73%) do not react, and in locations with a medium intensity of fear of crime, 54% of respondents do not react. On the other hand, in the locations with a high intensity of the fear of crime, only 33% of respondents do not use any means to protect themselves.

The comparison of recorded crime data with the fear of crime showed that there are areas where people feel afraid and the locations have higher crime rates. Nevertheless, some areas proved to be under/overrepresented in the perception of crime. The resulting percentage of hexagons classified in one of the three resulting categories is summarised in Table 4. The significantly higher percentage of hexagons classified as high recorded crime (without fear of crime) is caused mainly due to the higher number of recorded crime events compared to the fear of crime locations (more in Chapter 5—Conclusion). However, these hexagons cover only 1% of all hexagons covering the city. In the authors' understanding, it is crucial to identify also the areas where there is no or a weak link between recorded crime and the fear of crime, the way it has been done in this paper.

**Table 4.** Percentage of hexagons in three categories.

| Category | Fear of Personal Thefts × Thefts as Recorded Crime Incidents | Fear of Attack by a Drunk or Drug Addicted Person(s) × Personal Thefts |
|---|---|---|
| high-high × high-high (dark red hexagons) | 8.8% | 13.2% |
| High perception of crime (hexagons with blue hues) | 16.9% | 28.8% |
| High recorded crime (hexagons with green hues) | 74.3% | 57.9% |

The recorded crime data has its limitations as it does not contain misdemeanours that are often solved on the spot by the municipal police and not by the national police force. If misdemeanours entered the analysis, the resulting images could have been different and may have reflected more on the fear of crime in areas, such as the hostel described in Figure 5. The second limitation of the crime data is latent crime that is not covered by official police data. From the questionnaire, 28% of respondents in Ostrava experienced a crime incident during the last 12 months, while only a half of them reported it to the police. This makes the level of crime latency approximately 50%.

*Research limitations*

The main limitation of the proposed methodology is the point representation of a location where there is a higher fear of crime. There, people are afraid in not just one location, but over a larger area. This results in a smaller number of hexagons in the high-high × high-high class. The reason for using point representation in our study was discussed in chapter 3 (Methods and results). One possible solution is in the use of a kernel function that scatters the influence of a single point (in one hexagon) into surrounding hexagons within a defined threshold distance.

The second limitation is in the method used. The hexagons classified into clusters (high-high and low-low) represent the cores of larger areas and not the actual clusters. In contrast, hexagons classified as low-high and high-low clusters represent actual locations. This means that the resulting hexagons from high-high clusters are currently underestimated and define the cores of areas with a justified fear of crime.

The third limitation could be in the comparability of the results, where the authors are placing three years of recorded crime data on the same level as a ten week period of emotional mapping. The authors are aware that the perception of safety can change over time and place, and it is a context dependent experience [64]. However, in this research, the fear of crime is more of an experience-based perception, hence the responses to the emotional mapping can be traced back to the places for longer periods.

The fourth limitation can be the fact that the population density of the city has not been taken into account while exploring the recorded crime and perception data. As already mentioned in the Introduction, such data, that would help in the analysis, do not exist for the Czech Republic. The data available can address the points, but will not visualise the real-time population density in the city.

In order to improve the temporality of the responses, it has been suggested that researchers use mobile applications for the fear of crime mapping [65]. Although it may seem to be a promising approach, and one that has been tested in other emotional mapping scenarios [66], the authors are cautious about using mobile apps in the fear of crime mapping. The main obstacle is in the personal safety of the person doing the mapping and the mapping device (cell phone). To the authors' understanding, it is quite unlikely that respondents would take out of their pocket a device worth some hundreds of dollars to mark a spot on a map while feeling unsafe, in a dark park for example. On the other hand, it is definitely an interesting avenue for further research in the area of the perception of safety.

## 5. Conclusions

The research presented in this paper is based on an intensive, online map-based questionnaire with 1551 respondents from Ostrava, the Czech Republic. Over a ten week period, the respondents marked 3792 points associated with the fear of crime. This data was combined and compared with 257,381 incidents of recorded crime from the period of January 2015 to July 2018, which included all recorded crime offences that occurred in Ostrava and within a 5 km buffer area beyond its borders. The spatial analyses show that the recorded points are spatially clustered and that there are hotspots in the city. While closely studying the types of buildings and the natural environments of the fear of crime hotspots, the authors realised they are mostly locations with large anonymous crowds (train/bus stations), poor street lighting (parks or shadowy streets), and higher concentrations of homeless people and groups of people under the influence of drugs or alcohol (city centres, bars, clubs, etc.). The results from Ostrava show a comparison between the fear of crime and the recorded crime data acquired from the national police force.

Currently, our results are unable to be compared with any similar research in the Czech Republic as the only similar research was done in 1995 in the city of Příbram without any computers or GIS. The city of Ostrava is already working with our results/findings, especially in the areas that are monitored regarding urban safety from a long-term perspective. The aim of the authors is to repeat the research in one or two years in order to be able to compare the spatial and temporal changes in the fear of crime. This approach can discover new problematic locations and the city can implement preventive measures, such as CCTV cameras, more police patrols, greenery or public spaces changes, etc.

Further research may be focused on a deeper analysis of how people perceive safety in their neighbourhood compared to the rest of the city, or how their personal experience with crime affects their actions. The future research questions can include a comparison of the different locations or hexagons based on their land use or built environment. Are there locations that in general generate more fear or attract more crime? The results from such analysis can be compared with international research already done and described in the Introduction. Further exploration could also be focused on

reasons why some locations generate only fear, but not crime—Is it linked with almost 50% of crime being unreported? Are there stress factors that could have been omitted? Several questions from the questionnaires were not included in this paper as they require further analysis. The raw data from Ostrava is available for browsing at the research webpage [67].

**Author Contributions:** Conceptualization, Jiří Pánek, Lucie Macková and Igor Ivan; methodology, Jiří Pánek, Lucie Macková and Igor Ivan; formal analysis, Jiří Pánek, Lucie Macková and Igor Ivan; writing—original draft preparation, Jiří Pánek, Lucie Macková and Igor Ivan; writing—review and editing, Jiří Pánek, Lucie Macková and Igor Ivan; project administration, Igor Ivan; funding acquisition, Igor Ivan.

**Funding:** This research was financially supported by the Technology Agency of the Czech Republic (TAČR), grant number: TJ01000465, project: "Effective Methods of Identification, Assessment and Monitoring of Safety Risk Areas Using Spatial Micro-data".

**Acknowledgments:** The authors would like to thank the four anonymous reviewers for their work and suggestion regarding the paper. We would like to thank the City of Ostrava for their collaboration on our research as well as to all participants who filled in our questionnaire.

**Conflicts of Interest:** The authors declare no conflicts of interest.

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
