# Peer review of "Comparing Residents’ Fear of Crime with Recorded Crime Data—Case Study of Ostrava, Czech Republic"

_ijgi, doi:10.3390/ijgi8090401_

Round 1

Reviewer 1 Report

This manuscript presents a comparison of locations where residents of a city in the eastern Czech Republic were afraid of crime compared with where crimes were actually recorded. The study drew on police records of crime and reports of fear of crime collected through a participatory mapping exercise. 

The positive point of the manuscript is that the participatory exercise collected a fairly large amount of data from a good number of participants and that the authors procured access to crime data from official records. However, there are a number of problems with the manuscript that prevent me from being able to recommend its publication. At present I cannot really see what this manuscript contributes to our understanding of spatial patterns and causes of fear of crime in a general sense or to the methods for studying fear of crime. It does tell us something about spatial patterns of the phenomenon in Ostrava specifically, but I’m not sure that it does anything beyond that. I also have concerns with the way the authors used spatial statistical methods. I detail these problems below.

Inadequate use of the literature

There are a number of major works that the authors do not consider in their treatment of the literature, including a major monograph with an extended treatment of the spatiotemporal analysis of fear of crime. Overall, I think this shows an inadequate command of the literature in this area of scholarship and leads to claims that are not supported and (I think) also incorrect and not supported by the literature, like [from the abstract]: “… only limited research compare fear of crime with real crime data within the urban space.” I think there is actually quite a lot of work that shows in fact that fear of crime often exceeds actual crime.

The authors cite the broken window theory as a motivation to their work, but adopt an entirely uncritical consideration of the theory. There has been a very large amount of debate over the validity of this theory in criminological research circles, but here it is uncritically accepted and none of these points are acknowledged.

The authors should consult:

Doran, B. J., & Burgess, M. B. (2011). Putting fear of crime on the map: Investigating perceptions of crime using geographic information systems. Springer Science & Business Media.

Doran, B. J., & Lees, B. G. (2005). Investigating the spatiotemporal links between disorder, crime, and the fear of crime. The Professional Geographer, 57(1), 1-12.

Leitner, M. (Ed.). (2013). Crime modeling and mapping using geospatial technologies (Vol. 8). Springer Science & Business Media. — see particularly the Fuhrmann et al chapter, which directly compares fear of crime and crime data.

Pődör, A., & Dobos, M. (2014). Official crime statistics versus fear of crime of the citizens in a hungarian small town. GI_Forum, 272-275.

Alkimim, A., Clarke, K. C., & Oliveira, F. S. (2013). Fear, crime, and space: The case of Viçosa, Brazil. Applied Geography, 42, 124-132.

I find it also a bit curious that in their description of analysis of crime data, the authors cite only a large number of IGJI papers, as if they are either a) the most important contributions in this field or b) the only contributions to the field. 

Use of spatial statistics

It’s not clear to me that the authors really understand what they are doing in their use of spatial statistics. Any statistical tests should be chosen based on a question that the authors want to answer and a null hypothesis they are testing. What that is is not entirely clear. Is your aim to determine whether places where people are afraid of crime are also where they experience crime? If so, I do not see how Ripley’s K analysis assists with that.

Moreover, Ripley’s K is usually used to investigate clustering at a range of spatial scales, but the way the authors report their results implies that they are using it to say generally whether the spatial pattern contains clusters or not. That is not really the purpose of that statistic — what is measures is whether at a particular inter-event distance across all events in the dataset there are there are more pairs that are closer together than one would expect in a completely spatially random pattern. Moreover, the K function is falling out of favour because of the difficulty of its interpretation, in favour of the pair correlation function. See the O’Sullivan and Unwin book on geographical analysis for a discussion of relevant considerations.

Any consideration of crime data needs to account for variations in population density across the study area. It’s not a surprise that places with higher population might for example, have larger counts of crime events.

When aggregating your count data to hexbins, you do not explain why you chose that particular size. There is no sensitivity analysis that examines whether there are different findings when you aggregate to different scales. The modifiable areal unit problem is a well known issue with spatial data analysis and its existence demands that careful thought should go into any aggregation of data and one should test for whether there is evidence that the pattern found is simply an artefact of the aggregation method.

English language expression

While the manuscript is written in largely understandable English, there are many places where the writing could be made more precise and clear. For example, even in the first sentence: Understanding the response to the fear of crime — *whose* response? That of citizens? Policymakers? Police? Later in that paragraph “Moreover, it is important…” To what does ‘it’ refer? This is unclear and the rest of the sentence also does not clarify this. There are many more such examples throughout the paper — too many for me to list individually.  I recommend that the authors work with a native speaker to find and improve these areas where it’s not entirely clear what they mean.

Check also the use of ‘day time’. I think what is meant here is ‘time of day’. There is also a lot of use of the term ‘respective’ when there is not clear that there are two or more things being referred to — so the use of this term specifically should be checked by a native English speaker. Finally, the term ‘socially weak’ is not one that would be used in English. Do you mean poor people? People who are otherwise vulnerable?

Author Response

Reviewer 1

This manuscript presents a comparison of locations where residents of a city in the eastern Czech Republic were afraid of crime compared with where crimes were actually recorded. The study drew on police records of crime and reports of fear of crime collected through a participatory mapping exercise. 

The positive point of the manuscript is that the participatory exercise collected a fairly large amount of data from a good number of participants and that the authors procured access to crime data from official records. However, there are a number of problems with the manuscript that prevent me from being able to recommend its publication. At present I cannot really see what this manuscript contributes to our understanding of spatial patterns and causes of fear of crime in a general sense or to the methods for studying fear of crime. It does tell us something about spatial patterns of the phenomenon in Ostrava specifically, but I’m not sure that it does anything beyond that. I also have concerns with the way the authors used spatial statistical methods. I detail these problems below.

Inadequate use of the literature

There are a number of major works that the authors do not consider in their treatment of the literature, including a major monograph with an extended treatment of the spatiotemporal analysis of fear of crime. Overall, I think this shows an inadequate command of the literature in this area of scholarship and leads to claims that are not supported and (I think) also incorrect and not supported by the literature, like [from the abstract]: “… only limited research compare fear of crime with real crime data within the urban space.” I think there is actually quite a lot of work that shows in fact that fear of crime often exceeds actual crime.

The authors cite the broken window theory as a motivation to their work, but adopt an entirely uncritical consideration of the theory. There has been a very large amount of debate over the validity of this theory in criminological research circles, but here it is uncritically accepted and none of these points are acknowledged.

We do not build the whole paper around the broken windows theory, but we can see, that the lack of criticism may be alarming. We have added a short text to that, but the aim of the paper is not to describe different theories, but to analyse (by spatial statistics) the differences of recorded crime and fear of crime. 

The authors should consult:

Doran, B. J., & Burgess, M. B. (2011). Putting fear of crime on the map: Investigating perceptions of crime using geographic information systems. Springer Science & Business Media.

Doran, B. J., & Lees, B. G. (2005). Investigating the spatiotemporal links between disorder, crime, and the fear of crime. The Professional Geographer57(1), 1-12.

Leitner, M. (Ed.). (2013). Crime modeling and mapping using geospatial technologies (Vol. 8). Springer Science & Business Media. — see particularly the Fuhrmann et al chapter, which directly compares fear of crime and crime data.

Pődör, A., & Dobos, M. (2014). Official crime statistics versus fear of crime of the citizens in a hungarian small town. GI_Forum, 272-275.

Alkimim, A., Clarke, K. C., & Oliveira, F. S. (2013). Fear, crime, and space: The case of Viçosa, Brazil. Applied Geography42, 124-132.

Thank you for the resources, definitely worth exploring.

I find it also a bit curious that in their description of analysis of crime data, the authors cite only a large number of IGJI papers, as if they are either a) the most important contributions in this field or b) the only contributions to the field. 

True, this is unneeded over-representation of IJGI and this has been corrected.

Use of spatial statistics

It’s not clear to me that the authors really understand what they are doing in their use of spatial statistics. Any statistical tests should be chosen based on a question that the authors want to answer and a null hypothesis they are testing. What that is is not entirely clear. Is your aim to determine whether places where people are afraid of crime are also where they experience crime?

The aim of the paper is to investigate whether the fear of crime occurs in the areas where the recorded crime and vice versa.

If so, I do not see how Ripley’s K analysis assists with that.

Moreover, Ripley’s K is usually used to investigate clustering at a range of spatial scales, but the way the authors report their results implies that they are using it to say generally whether the spatial pattern contains clusters or not. That is not really the purpose of that statistic — what is measures is whether at a particular inter-event distance across all events in the dataset there are there are more pairs that are closer together than one would expect in a completely spatially random pattern. Moreover, the K function is falling out of favour because of the difficulty of its interpretation, in favour of the pair correlation function. See the O’Sullivan and Unwin book on geographical analysis for a discussion of relevant considerations.

After reading your and other reviews, we decided to remove K function part at all, because it does not say anything new. Moran’s I that is used later proved that the distribution tends to be clustered in space (see Table 2).

Any consideration of crime data needs to account for variations in population density across the study area. It’s not a surprise that places with higher population might for example, have larger counts of crime events.

We definitely agree it is not surprising. Unfortunately, population data (of present people not from the census) is not accessible for us. This is why we decided to compare spatial distributions of two phenomenons (fear of crime and recorded crime) instead of focusing just on the spatial distribution of one phenomenon.

Data about real population density do not exist for the Czech Republic. There are data of where people live, but this is a density of “sleeping” people, and it is quite different from real-time population density in the city. Therefore it made no sense to anayse crime data (both recorded and fear of crime) with population density.

The second reason is, that the density data are only available on district level that is absolutely not comparable with the hexagon we have used for our aggregations.

When aggregating your count data to hexbins, you do not explain why you chose that particular size. There is no sensitivity analysis that examines whether there are different findings when you aggregate to different scales. The modifiable areal unit problem is a well known issue with spatial data analysis and its existence demands that careful thought should go into any aggregation of data and one should test for whether there is evidence that the pattern found is simply an artefact of the aggregation method.

We added this part in the chapter Method and results where we checked different lengths of hexagon sides and compared resulting Global Moran’s I and number of hexagons.

English language expression

While the manuscript is written in largely understandable English, there are many places where the writing could be made more precise and clear. For example, even in the first sentence: Understanding the response to the fear of crime — *whose* response? That of citizens? Policymakers? Police? Later in that paragraph “Moreover, it is important…” To what does ‘it’ refer? This is unclear and the rest of the sentence also does not clarify this. There are many more such examples throughout the paper — too many for me to list individually.  I recommend that the authors work with a native speaker to find and improve these areas where it’s not entirely clear what they mean.

Check also the use of ‘day time’. I think what is meant here is ‘time of day’. There is also a lot of use of the term ‘respective’ when there is not clear that there are two or more things being referred to — so the use of this term specifically should be checked by a native English speaker. Finally, the term ‘socially weak’ is not one that would be used in English. Do you mean poor people? People who are otherwise vulnerable?

The whole paper went through language editing by a professional proof-reader (British) with specialisation in geography texts.

Reviewer 2 Report

I enjoyed reading this paper and it has the potential to be a useful contribution to the literature.

However, there are a number of points that need addressing before the paper can be considered publishable.

At times the word selection or sentence structure is difficult to read or doesn’t flow well – I expect as the authors first language is not English. Some support here would help flow of text. There are also some grammatical errors. I would avoid using the phrase ‘real crime’. All measurements of crime are effectively an estimate and we don’t know the true/real level. When you are using police recorded crime data why not keep with ‘recorded crime’ as opposed to ‘real crime’. This should be changed throughout. Fear of crime itself is a socially constructed concept and there is debate in the literature as to what this means. This needs to be reflected in both the literature (define fear of crime for this paper) and in the limitations section - considering the use of the app to capture fear of crime data – and the individual subjectivity of this concept. It is not sufficient in the justification to suggest the case study was selected, because it was part of a wider project. Whilst this supports the argument it cannot be the only reason. Beyond this why was the case study area appropriate for the study you wished to do. In section 2 you discuss the impact of mixed land use for reducing crime. There is mixed evidence here so take care with this argument. I have read contradictory findings in different study areas. Page 3 – be careful with your use of the word attractor. This is a specific concept used by the Brantinghams around crime attractors/crime generators and each have a specific meaning. If you refer to this how do you know this is an attractor rather than a generator. Or use a different word. Section 3 is a more appropriate justification for the study (see point 4) Page 4 – why did you use Ripleys as opposed to other point data clustering technique (eg GI* identifies statistically significant hotspots). The analysis does not need re-doing but the technique needs justification. Page 6- You now use registered crime data. Keep your language consistent. Prefer recorded. Page 6- why have you selected these five crime types – violent crime/burglaries/extremism/personal theft/car robbery – do previous studies suggest they are likely to be associated with fear of crime. How might fear of crime vary by differing crime types. Page 7 Your high-high, or low low clusters etc need more descriptive information on how and why they were calculated. What was the reasons for looking at recorded crime and fear of crime both within hexagon and in surrounding six hexagons. Would it be better to do analysis on hexagons alone first, and then on hexagons and nearby hexagons. Is the figure an average of the six surrounding hexagons (high mean). What is impact of several hexagons with 0 values. Have you used standard deviations or other. What constitutes a high and a low mean. This should then be used to support discussion of Table 2. There needs to be more discussion of justification of this approach. The limitations of this method (in creating Table 1 and 2 and associated figures) needs to be included in limitations section. Page 8 – avoid using words such as problematic as this is journalistic/vague. I wonder if the data on responses to crime is actually a separate paper. This would allow you to spend more time discussing the methodology, interpreting the findings, and also considering the findings in context of the limitations of the study. Page 11- you discuss areas with high levels of recorded crime and low levels of fear – what might the reasons for this be – limitations of data/method or other. What might explanations for this be and how should this be explored in future research. What are the policy implications of this. In the conclusion you state you are comparing 10 weeks data with 3 years data. Is this comparable. What are the limitations of this. A limitation section needs to be added to discussion section. A major limitation is that fear of crime whilst spatially analysed is not analysed by time of day or day of week. This is one of the advantages of using app based fear of crime work – in that you can ask more specifically about fear at a given situation. This might be an area of future research you can highlight – and perhaps should also be considered in your literature discussion. For example see work by Solymosi on fear of crime: Solymosi, R., Fujiyama, T. & Bowers, K., 2015, Mapping fear of crime as a context-dependent everyday experience that varies in space and time. Legal and Criminological Psychology. 20, 2 Are there policy recommendations from this study.

Author Response

I enjoyed reading this paper and it has the potential to be a useful contribution to the literature. However, there are a number of points that need addressing before the paper can be considered publishable.

Thank you for your review, your comments were highly valuable and definitely helped us to shape the paper into a more solid presentation of our research.

At times the word selection or sentence structure is difficult to read or doesn’t flow well – I expect as the authors first language is not English. Some support here would help flow of text. There are also some grammatical errors.

The whole paper went through language editing by a professional proof-reader (British) with specialisation in geography texts.

I would avoid using the phrase ‘real crime’. All measurements of crime are effectively an estimate and we don’t know the true/real level. When you are using police recorded crime data why not keep with ‘recorded crime’ as opposed to ‘real crime’. This should be changed throughout.

Thank you for this suggestion, we changed the wording in the whole paper.

Fear of crime itself is a socially constructed concept and there is debate in the literature as to what this means. This needs to be reflected in both the literature (define fear of crime for this paper)

We have added more definitions of fear of crime in the Background section and expanded our understanding of the concept in the paper.

and in the limitations section - considering the use of the app to capture fear of crime data –

We are not in favour of using apps for fear of crime, but we elaborated on this in the limitations section. Thank you for the suggestion, but we are not sure if we are ready for such a change in our approach.

and the individual subjectivity of this concept.

This has been shortly discussed in the Background section.

It is not sufficient in the justification to suggest the case study was selected, because it was part of a wider project. Whilst this supports the argument it cannot be the only reason. Beyond this why was the case study area appropriate for the study you wished to do.

We have added the whole paragraph on case study location selection to the Introduction, where we feel it fits better. Although our justification may not be optimal, the truth is, we were lucky to get the data (registered crime) – the first time any Czech university received data on registered crime in the Czech Republic. We were not in the position to be too picky – it took over a year of negotiations with the Police Presidium to get the data.

In section 2 you discuss the impact of mixed land use for reducing crime. There is mixed evidence here so take care with this argument. I have read contradictory findings in different study areas.

We have both sides of the argument in the Background section:

Research also shows that specific types of crime correlate with particular characteristics of urban spaces Therefore, it is important to consider these aspects when planning the urban design. For example, Stankevice et al. demonstrate that specialised areas and greenery in dense residential areas contribute to crime prevention. However, if these areas are combined with local centres, commercial or industrial areas, they become even more attractive to criminals. Similarly, Hillier and Sahbaz argued that mixed-use street segments are relatively safe in the majority of cases and that increased residential population neutralises the risks associated with sparse residence in mixed-use areas. Moreover, the lack of residents in public spaces and the discontinuity of public spaces increase crime because of the lack of oversight by other residents. This is in line with the pioneering findings of Jacobs who claimed that urban spaces with mixed-use lead to less crime because they provide more natural surveillance. Studies also show that crime hotspots can change over time, but also discuss spatial periodicity of the crime itself.

Page 3 – be careful with your use of the word attractor. This is a specific concept used by the Brantinghams around crime attractors/crime generators and each have a specific meaning. If you refer to this how do you know this is an attractor rather than a generator. Or use a different word.

True, thank you, we changed the wording accordingly.

Section 3 is a more appropriate justification for the study (see point 4)

Yes, and we moved it directly to the Introduction + we added few more lines to it.

Page 4 – why did you use Ripleys as opposed to other point data clustering technique (eg GI* identifies statistically significant hotspots). The analysis does not need re-doing but the technique needs justification.

After reading your and other reviews, we decided to remove k-function part at all, because it does not say anything new. Moran’s I that is used later proved that the distribution tends to be clustered in space (see Table 2).

Page 6- You now use registered crime data. Keep your language consistent. Prefer recorded.

Yes, changed as suggested, thank you!

Page 6- why have you selected these five crime types – violent crime/burglaries/extremism/personal theft/car robbery – do previous studies suggest they are likely to be associated with fear of crime. How might fear of crime vary by differing crime types.

We have added the explanation in the text like this one: Five categories/classes of crime offences (violent crime, burglaries, extremism, personal theft and car robbery) were selected based on several reasons. Firstly, these crime offences were recommended by police and Ostrava authorities. Second, the total number of incidents in each of selected category/class is high enough for further spatial analysis, compared to many other categories/classes with a limited number of incidents (e.g. sexual offences, arsons, frauds). Third, the spatial distribution of events is not as much influenced by external factors as some other categories/classes (e.g. traffic accidents, driving offences). Fourth, selected categories/classes can be compared with categories defined in fear of crime data.

Page 7 Your high-high, or low low clusters etc need more descriptive information on how and why they were calculated.

We extended the descriptive text in the chapter on Method and Results and also added a new map focusing only on one aspect (fear of crime) as the first product for main results of the paper – combined cluster map.

We extended and modified the text explaining the methodology and resulting categories, also added one reference to Anselin (1995) where the methodology is explained in detail.

What was the reasons for looking at recorded crime and fear of crime both within hexagon and in surrounding six hexagons. Would it be better to do analysis on hexagons alone first, and then on hexagons and nearby hexagons.

We decided to use both crimes within hexagon and in surrounding six hexagons because the point localisation has its limitations – a person is feeling fear in a larger area but can use only one point to highlight it. The used methodology (explained in the Methods and results chapter and equation) is comparing values in a single unit with overall mean together with its surrounding that is defined as adjacent hexagons in the paper.

Is the figure an average of the six surrounding hexagons (high mean).

The resulting combined cluster maps (fig 5 and 6) are combinations of two different simple cluster maps (one in fig. 4) and only hexagons with a combination containing at least one high-high cluster were displayed in the final map. 

Even in calculation there is not computing an average of six surrounding hexagons but overall mean (the text was modified to be more comprehensive). As it is described in the equation, it is a product of two differences (value in the hexagon i and overall mean and hexagon y and overall mean). For each hexagon, there will be non-zero results of these products only for 6 adjacent hexagons (wij is one).

What is impact of several hexagons with 0 values. Have you used standard deviations or other.

These zero values are decreasing the overall mean value that is gradually compared with values in hexagons.

What constitutes a high and a low mean.

This was modified in the text.

This should then be used to support discussion of Table 2.

The text, as well as both maps, were revised.

There needs to be more discussion of justification of this approach. The limitations of this method (in creating Table 1 and 2 and associated figures) needs to be included in limitations section.

Extended limitations of this method are described in chapter Discussion.

Page 8 – avoid using words such as problematic as this is journalistic/vague.

Done

I wonder if the data on responses to crime is actually a separate paper. This would allow you to spend more time discussing the methodology, interpreting the findings, and also considering the findings in context of the limitations of the study.

This is an interesting suggestion, we will keep it in mind while working further on the project.

Page 11- you discuss areas with high levels of recorded crime and low levels of fear – what might the reasons for this be – limitations of data/method or other. What might explanations for this be and how should this be explored in future research. What are the policy implications of this.

It can be both the limitation of data or method used as well as the type of crimes happening in certain places. In can be also that the crimes are relatively “new”, hence the genius loci did not have enough time to be created. This question currently cannot be answered.

In the conclusion you state you are comparing 10 weeks data with 3 years data. Is this comparable. What are the limitations of this.

Limitations subsection was added to the Discussion section.

A limitation section needs to be added to discussion section. A major limitation is that fear of crime whilst spatially analysed is not analysed by time of day or day of week.

Limitations subsection was added to the Discussion section.

This is one of the advantages of using app based fear of crime work – in that you can ask more specifically about fear at a given situation. This might be an area of future research you can highlight – and perhaps should also be considered in your literature discussion.

Limitations subsection was added to the Discussion section, where using a mobile app was also discussed.

For example see work by Solymosi on fear of crime: Solymosi, R., Fujiyama, T. & Bowers, K., 2015, Mapping fear of crime as a context-dependent everyday experience that varies in space and time. Legal and Criminological Psychology. 20, 2

Very interesting paper, thank you!

Are there policy recommendations from this study.

As stated in the discussion section, respondents feel that the safety situation in Ostrava is getting worse and at the same time, they see no activities of crime prevention by the city. The policy recommendations can be more in the awareness-raising area.

Reviewer 3 Report

This is an interesting paper that investigates the spatial patterns of crime in Ostrava, Czech by comparing data on fear of crime and police reported incidents. While it is well-written, I have found a number of critical issues that should be addressed by the authors before it can be accepted for publication.

The place names in some of the maps including Figure 1, 3, 4, and 5 are barely legible, which made it impossible for readers to get to know the locations and places that have been referred in the text.  Descriptive statistics for both registered crime and fear of crime should be reported in tables. Table 1 reported values of Moran's I but failed to include z scores and p-values that allow readers to know the significance of each Moran's I value. The interpretations of local Moran's I values are questionable. Results of high-high from local Moran's I analysis indicate hot spots of nearby spatial units that have similarly high or low values instead of just high values.  Authors should clearly describe how global and local spatial statistical methods (i.e., Moran's I) are used to investigate the clustering patterns of crime.... in the Data and Methods section. instead of using Moran'I, it would be more reasonable to use global and local General G statistics that enable users to identify clusters of high or low values among spatial data. Moran's I diagnosis won't differeiate whether clusters are caused by high-high or low-low observations that are neighbors.  The introduction section and literature review should be enhanced to include more existing research that analyzes fear of crime and police reported crime, for example, Weatherford (1995) and Curiel and Bisop (2018).  The conclusion section needs to be improved to discuss limitations of the current research. For example, one limitation of using police recorded incidents as reference is that some crimes have never been reported to the police. 

References -

Curiel, R.P. and Bishop, S.R., 2018. Fear of crime: the impact of different distributions of victimisation. Palgrave Communications4(1), p.46

Grogger J, Weatherford S (1995) Crime, policing and the perception of neighborhood safety. Political Geogr 14(6/7):521–541

Author Response

This is an interesting paper that investigates the spatial patterns of crime in Ostrava, Czech by comparing data on fear of crime and police reported incidents. While it is well-written, I have found a number of critical issues that should be addressed by the authors before it can be accepted for publication.

The place names in some of the maps including Figure 1, 3, 4, and 5 are barely legible, which made it impossible for readers to get to know the locations and places that have been referred in the text. 

Thank you for notifying us, text labels were added into the maps where it was relevant to the text

Descriptive statistics for both registered crime and fear of crime should be reported in tables.

We consider whether or not to add such table, but we prefer not to add a table with descriptive statistics while it is not directly relevant to the aim of the paper and not surprising at all. We did it during the analytical part. Majority of hexagons have no event, non-zero hexagons have on average from 1.2 to 2.5 events, but the median is 1 for all types of crime. Distribution is highly left-skewed and peaked.

Table 1 reported values of Moran's I but failed to include z scores and p-values that allow readers to know the significance of each Moran's I value.

We added this information

The interpretations of local Moran's I values are questionable. Results of high-high from local Moran's I analysis indicate hot spots of nearby spatial units that have similarly high or low values instead of just high values.  Authors should clearly describe how global and local spatial statistical methods (i.e., Moran's I) are used to investigate the clustering patterns of crime.... in the Data and Methods section. instead of using Moran'I, it would be more reasonable to use global and local General G statistics that enable users to identify clusters of high or low values among spatial data. Moran's I diagnosis won't differeiate whether clusters are caused by high-high or low-low observations that are neighbors.

Thank you for this note. Global Moran's I was used to analyse if and how much the events are clustered in the city without specifying if high-high or low-low clusters prevail. We ran the Global General G what proved the prevalence of high-high clusters, but we do not want to use another index in the paper as the main goal is not to say whether or not the events are clustered (all know that they are). We want to say if clusters of fear of crime data are at the same locations as clusters of recorded crime data and for this local Moran's I is applicable. Local Moran' s I can differentiate if high values or low values are clustered (classify hexagons into 5 categories - not significant, high-high, low-low, high-low and low-high). We modified the text to highlight the above-written explanation.

The introduction section and literature review should be enhanced to include more existing research that analyzes fear of crime and police reported crime, for example, Weatherford (1995) and Curiel and Bisop (2018). 

The introduction was extended by new literature in various parts.

The conclusion section needs to be improved to discuss limitations of the current research. For example, one limitation of using police recorded incidents as reference is that some crimes have never been reported to the police. 

we extended the limitation of police data in the discussion part

References -                                                        

Curiel, R.P. and Bishop, S.R., 2018. Fear of crime: the impact of different distributions of victimisation. Palgrave Communications4(1), p.46

Grogger J, Weatherford S (1995) Crime, policing and the perception of neighborhood safety. Political Geogr 14(6/7):521–541

Reviewer 4 Report

This is an interesting paper that holds potential however there are grammatical errors throughout. My recommendation is that the paper is adjusted, thoroughly proof-read and re-submitted. Some of these errors, alongside some other considerations, are listed below:

Page 1

With regards to your title and the rest of the paper, would ‘‘recorded crime data’’ perhaps be a better choice of words than ‘‘real crime data’’? Smaller points about the abstract: perhaps remove the word ‘‘emerging’’? It is established and not emerging – ‘‘limited research compares’’ – ‘‘within urban spaces’’ – ‘‘this paper presents the results of an intensive online map’’ - ‘‘fear of crime within a ten week period’’. The first sentence of your introduction is confusing and can be removed. ‘‘one of the essential concepts of Criminology’’. ‘‘how likely it is that one will be a victim of crime’’ ‘‘it is important for policymakers’’. ‘‘in order to explore the attributes of various places in a selected city’’ ‘‘GIS are well-positioned to integrate data on people’s perceptions’’

Page 2

‘‘Our research focuses on the Czech city of Ostrava’’ ‘‘The involvement in the project is linked with prior collaboration with this participating university as well’’. ‘‘have never been compared in the Czech Republic’’. ‘‘Fear of crime emerged as a standalone concept in the 1960s’’ – although you might want to think about removing this statement because it does not add anything. ‘‘crime hotspots can change over time’’

Page 3

‘‘and that public transportation is an important, albeit neglected, dimension of policies targeting quality of life’’. ‘‘the open spaces and outdoor recreational areas facilitate larceny theft’’.

Page 4

‘‘questions regarding the time of day respondents do not feel safe’’. ‘‘this paper analyses only the spatial part of the questionnaire’’. ‘‘there are no significant differences in answers of university students and the general population. Whether this is the case in Ostrava has not been proved yet’’.

Page 5

What is meant by ‘‘socially weak inhabitants’’? ‘‘this may have been caused mainly by cheap local hostels and localities’’.

Author Response

Reviewer 4

This is an interesting paper that holds potential however there are grammatical errors throughout. My recommendation is that the paper is adjusted, thoroughly proof-read and re-submitted. Some of these errors, alongside some other considerations, are listed below:

Thank you very much for your comments. The whole paper went through language editing by professional proof-reader (British) with specialisation on geography texts. We hope our edits improved the quality of the paper.

Page 1

With regards to your title and the rest of the paper, would ‘‘recorded crime data’’ perhaps be a better choice of words than ‘‘real crime data’’? Smaller points about the abstract: perhaps remove the word ‘‘emerging’’? It is established and not emerging – ‘‘limited research compares’’ – ‘‘within urban spaces’’ – ‘‘this paper presents the results of an intensive online map’’ - ‘‘fear of crime within a ten week period’’. The first sentence of your introduction is confusing and can be removed. ‘‘one of the essential concepts of Criminology’’. ‘‘how likely it is that one will be a victim of crime’’ ‘‘it is important for policymakers’’. ‘‘in order to explore the attributes of various places in selected city’’ ‘‘GIS are well-positioned to integrate data on people’s perceptions’’

Page 2

‘‘Our research focuses on the Czech city of Ostrava’’ ‘‘The involvement in the project is linked with prior collaboration with this participating university as well’’. ‘‘have never been compared in the Czech Republic’’. ‘‘Fear of crime emerged as a standalone concept in the 1960s’’ – although you might want to think about removing this statement because it does not add anything. ‘‘crime hotspots can change over time’’

Page 3

‘‘and that public transportation is an important, albeit neglected, dimension of policies targeting quality of life’’. ‘‘the open spaces and outdoor recreational areas facilitate larceny theft’’.

Page 4

‘‘questions regarding the time of day respondents do not feel safe’’. ‘‘this paper analyses only the spatial part of the questionnaire’’. ‘‘there are no significant differences in answers of university students and the general population. Whether this is the case in Ostrava has not been proved yet’’.

Page 5

What is meant by ‘‘socially weak inhabitants’’? ‘‘this may have been caused mainly by cheap local hostels and localities’’. 

Round 2

Reviewer 1 Report

The revision is a definite improvement on the original submission. The authors have improved their paper using the feedback from referees though there are still some issues with the methodology and the writing.

I am wondering why the authors did not also use a bivariate LISA statistic for the analysis reported in Figures 4 & 5?

https://geodacenter.github.io/workbook/6b_local_adv/lab6b.html#bivariate-local-morans-i

It's also not clear to me why the low-low and low-high areas are omitted (ie not symbolised) from Figure 3. Areas of low fear of crime should also be of interest.

I think the fact that you do not have crime rates but rather crime counts should be discussed as a limitation. Population distribution also provides context for interpreting unexpectedly low or high numbers of crimes, so not having this challenges your ability to provide this context.

Finally, there is mention of the K-function in the conclusion section, so I think the authors missed removing that in their revision.

Author Response

The revision is a definite improvement on the original submission. The authors have improved their paper using the feedback from referees though there are still some issues with the methodology and the writing.

Thank you, we did our best and the reviews really helped us to shape the paper!

I am wondering why the authors did not also use a bivariate LISA statistic for the analysis reported in Figures 4 & 5?

It is definitely a possibility of how to further analyse the data. We have considered it but decided not to use it in the paper because this method does not analyse the correlation between both variables at the same location (hexagon) but the first variable in the hexagon with the second variable in the surroundings. We preferred to analyse the correlation in the hexagon for both variables.

https://geodacenter.github.io/workbook/6b_local_adv/lab6b.html#bivariate-local-morans-i

It's also not clear to me why the low-low and low-high areas are omitted (ie not symbolised) from Figure 3. Areas of low fear of crime should also be of interest.

We have omitted other categories except for high-high and high-low categories just to make the map more comprehensible. The problem with the low-low cluster is that hexagons from this category are located in areas where crime does not happen or it is almost impossible for it to happen (industrial complexes, greenery, rivers, agricultural land etc.). Ostrava is a typical polycentric city with large non-urbanised areas among its three centres and with small surrounding villages around it but still within administrative boundaries of the city. Thus low-low cluster mainly locates areas with no crime at all.

I think the fact that you do not have crime rates but rather crime counts should be discussed as a limitation. Population distribution also provides context for interpreting unexpectedly low or high numbers of crimes, so not having these challenges your ability to provide this context.

Thank you, we agree that the fact we do not have population/crime rates data is a limitation but as it is emphasized in the paper, the main goal of the paper is not to locate areas with high crime (it is hard without the population data) but to compare locations with high recorded crime intensity and fear of crime intensity. We have added this into limitations of the research as one paragraph.

Finally, there is mention of the K-function in the conclusion section, so I think the authors missed removing that in their revision.

You are right, sorry about that, the section was deleted.

Reviewer 3 Report

The revised manuscript is in much better shape.

One minor issue is that the authors mentioned "Chapter 3" (Research limitations, line 4), which is is confusing. I guess it means the Data and methods section. 

Another issue is inconsistency in format of references. Some articles are input with the first letter in upper case but some are in lower case.

Author Response

The revised manuscript is in much better shape.

Thank you, we did our best and the reviews really helped us to shape the paper!

One minor issue is that the authors mentioned "Chapter 3" (Research limitations, line 4), which is is confusing. I guess it means the Data and methods section. 

Thank you for this point, we added the name of the chapter to clarify the point.

Another issue is inconsistency in format of references. Some articles are input with the first letter in upper case but some are in lower case.

Thank you, but we have decided to keep it the way it is as we use the original titles – different style comes from different journals but you are right, it looks a bit inconsistent.

Reviewer 4 Report

Comparing residents’ fear of crime with real crime data – case study Ostrava, Czech Republic

Thank you for your resubmission. Improvements have been made to the reporting and presentation of your research findings. However, there are still some limitations. There needs to be greater emphasis upon the unique contribution that this paper offers. After reading this paper, I was left with several questions that I feel need to be addressed:

What is the link between fear of crime and recorded crime? Why does it matter? If the places where individuals are fearful are completely different to crime hotspots, is this an issue? If they are similar, is this informative? Can these findings be built upon in the future and if so, how? (Why would this be necessary? Would there be policy implications for research of this nature?) (linked to the previous point) How do these findings compare to other places? Are they different, the same or is it unclear? Currently, it seems just an in-depth description of Ostrava is presented.

Furthermore, in the discussion section there is analysis of findings (for example, relating to gender). Can these be analysed and explored further within the analysis and findings section?

Author Response

Thank you for your resubmission. Improvements have been made to the reporting and presentation of your research findings.

Thank you, we did our best and the reviews really helped us to shape the paper!

However, there are still some limitations. There needs to be greater emphasis upon the unique contribution that this paper offers. After reading this paper, I was left with several questions that I feel need to be addressed:

What is the link between fear of crime and recorded crime?

The resulting percentage of hexagons classified in one of three resulting categories is summarised in Table 4. The significantly higher percentage of hexagons classified as a high recorded crime (without fear of crime) is caused mainly due to the higher number of recorded crime events compared to fear of crime locations (more in Chapter 5 – Conclusion). However, these hexagons cover only 1% of all hexagons covering the city.

Why does it matter? If the places where individuals are fearful are completely different to crime hotspots, is this an issue? If they are similar, is this informative?

In our understanding it is crucial to identify the areas where there is no or weak link between recorded crime and fear of crime (this paper) and then to analyse further (probably in future research and it has also been added to the future outlook section) why this happens - is it linked with almost 50% of crime being unreported? Are there stress factors that could have been omitted? 

Can these findings be built upon in the future and if so, how? (Why would this be necessary? Would there be policy implications for research of this nature?) (linked to the previous point)

The city is already working with our results/findings especially in the areas that are monitored regarding urban safety from a long-term perspective. We plan to repeat our research in one or two years in order to be able to compare the spatial and temporal changes in fear of crime. Such an approach can discover new areas and the city can implement preventive measures – such as CCTV cameras, more police patrols, greenery or public spaces changes, etc.

How do these findings compare to other places?  Are they different, the same or is it unclear? Currently, it seems just an in-depth description of Ostrava is presented.

Currently, we are unable to compare our results with any similar research in the Czech Republic, as the only “similar” research was done in 1995 in the city of Příbram and this was without any computers, nor GIS.

Regarding the international research, we believe that our findings fit into the literature confirming results about locations that most frequently attract fear, but there is still room for more investigations regarding the links between recorded crime and fear of crime. 

Furthermore, in the discussion section there is analysis of findings (for example, relating to gender). Can these be analysed and explored further within the analysis and findings section?

This is an interesting suggestion, but we are afraid that this would mean another full paper and adding just one small portion into this paper would not benefit the paper or the reader.

Round 3

Reviewer 1 Report

I have no further comments about the manuscript and I think it is now acceptable.

Reviewer 4 Report

Thank you for your comments. I think this article would be improved if you incorporated some of these responses into the main body of the text to make it clearer to the reader (especially the points about when similar research was conducted and the plan to repeat the research in one or two years). 

With regards to the last point about the exploration of gender within the discussion section. If there is not space to address this concept sufficiently in the findings section then it should be removed altogether; my understanding is that a discussion section should round up and build upon your argument. There should not be a presentation of concepts that were not addressed. 

Author Response

Thank you for your comments. I think this article would be improved if you incorporated some of these responses into the main body of the text to make it clearer to the reader (especially the points about when similar research was conducted and the plan to repeat the research in one or two years). 

Thank you very much for your suggestions, we incorporated most of our answers to the text – in various places – abstract, discussion and conclusions.

With regards to the last point about the exploration of gender within the discussion section. If there is not space to address this concept sufficiently in the findings section then it should be removed altogether; my understanding is that a discussion section should round up and build upon your argument. There should not be a presentation of concepts that were not addressed. 

We understand your point, but we decided to keep the information in the text as gender and other information about our research are not just concepts, but also parts of our results, which we addressed – via females not reporting their crimes more, etc. Taking this part out of the paper would mean, taking also the section about future outlook out and we definitely want to keep it in the paper. We hope the reviewer will understand our motivation.